# Thalamic Neuron Resilience during Osmotic Demyelination Syndrome (ODS) Is Revealed by Primary Cilium Outgrowth and ADP-ribosylation factor-like protein 13B Labeling in Axon Initial Segment

**DOI:** 10.3390/ijms242216448

**Published:** 2023-11-17

**Authors:** Jacques Gilloteaux, Kathleen De Swert, Valérie Suain, Charles Nicaise

**Affiliations:** 1URPhyM, NARILIS, Université de Namur, Rue de Bruxelles 61, B-5000 Namur, Belgium; jacques.gilloteaux@unamur.be (J.G.); kathleen.deswert@unamur.be (K.D.S.); 2Department of Anatomical Sciences, St George’s University School of Medicine, Newcastle upon Tyne NE1 JG8, UK; 3Laboratoire d’Histologie Générale, Université Libre de Bruxelles, Route de Lennik 808, B-1070 Bruxelles, Belgium; valerie.suain@ulb.be

**Keywords:** ARL13B, primary cilium, thalamus, neuron, osmotic demyelination syndrome, axon initial segment

## Abstract

A murine osmotic demyelinating syndrome (ODS) model was developed through chronic hyponatremia, induced by desmopressin subcutaneous implants, followed by precipitous sodium restoration. The thalamic ventral posterolateral (VPL) and ventral posteromedial (VPM) relay nuclei were the most demyelinated regions where neuroglial damage could be evidenced without immune response. This report showed that following chronic hyponatremia, 12 h and 48 h time lapses after rebalancing osmolarity, amid the ODS-degraded outskirts, some resilient neuronal cell bodies built up primary cilium and axon hillock regions that extended into axon initial segments (AIS) where ADP-ribosylation factor-like protein 13B (ARL13B)-immunolabeled rod-like shape content was revealed. These AIS-labeled shaft lengths appeared proportional with the distance of neuronal cell bodies away from the ODS damaged epicenter and time lapses after correction of hyponatremia. Fine structure examination verified these neuron abundant transcriptions and translation regions marked by the ARL13B labeling associated with cell neurotubules and their complex cytoskeletal macromolecular architecture. This necessitated energetic transport to organize and restore those AIS away from the damaged ODS core demyelinated zone in the murine model. These labeled structures could substantiate how thalamic neuron resilience occurred as possible steps of a healing course out of ODS.

## 1. Introduction

Among several CNS demyelinating defects, osmotic demyelinating syndrome or ODS is a serious neurological condition often consequent upon perturbations of the serum (Na^+^) homeostatic levels, such as those iatrogenically induced after an inappropriate management of chronic hyponatremia. This syndrome was first described in alcoholic patients by Adams and collaborators [1,2,3], as well as other authors [4,5,6,7,8]. It was described as a non-inflammatory neuropathologic condition that mainly encompassed regional CNS demyelination [4,5,6,7,8]. Moreover, ODS has been found accompanied by a broad clinical symptomatology, caused by clinical situations other than alcoholism that can involve slight confusion, disorientation, deafness, memory loss to seizure, paresis, unresponsiveness and eventually coma and death, depending on the degree and regional size of myelin loss in the pons as the ‘central pontine myelinolysis’ (CPM) and ‘extrapontine myelin’ (EPM) lesions [9,10,11,12,13,14,15,16,17,18,19,20,21,22,23,24,25,26,27,28,29,30,31,32,33,34,35,36,37,38,39,40,41,42,43,44]. It seemed that EPM lesions can appear before any CPM ones [30,31,37,38,43,44] and EPM cases would be more frequent than CPM cases. Additionally, the percentage of patients diagnosed with ODS has recently increased due to more preventative care being applied in clinical settings, especially with the more frequent usage of radiology diagnostic techniques, mainly magnetic resonance imagery (MRI) [9,10,11,12,13,14,15,16,17,18,19,20,21,22,23,24,25,26,27,28,29,30,31,32,33,34,35,36,37,38,39,40,41,42,43,44]. As verified from alcoholism [1,2,3,4,5,6,7,8,9,10,11,12,13,14,15,16,17,18,19,20,21,22,24], the ODS neuropathologic syndrome has been usually caused by a hasty adjustment of a chronic deficiency of the homeostatic sodium serum level [12,16,19,24,29,31,35,45,46,47,48]. However, this is insofar as the outcomes of either CPM or EPM have been reported as in 1999 [48], because both electrophysiological and MRI findings cannot predict the clinical outcome for patients with cerebral myelinolysis. 

The early human ODS histopathology [1,2,3,4,5,6,7,8] described clear nerve fibers demyelinated where myelin sheets were dilatated, vacuolated and fragmented while axon extensions seemed preserved, along with damage to neurons and astrocytes. Astrogliosis was evident and the oligodendrocyte population was decimated while numerous microglial cells, loaded with lipids and other neuropile captures, were observed in the cerebral lesioned regions and the ultrastructure of the few human cases examined confirmed the light microscopy reports; therein, blood vessel endothelium maintained most of their tight junctions and astrocytes showed swollen end-feet and other neuroglial cell degradations [1,2,3,4,5,6,7,8].

Basic experimental research into ODS has been carried out in small mammals [4,31,32,33,34,35,36,37,38,39,40,41,42,43,44,45,46,47,48,49,50,51,52,53,54,55,56,57,58,59,60,61,62,63,64,65,66,67,68,69], including mice in our laboratories [49,50,51,52,53,54,55,56,57,58,59,60,61,62,63,64,65,66,67,68,69]. Our laboratories contributed to showing that ODS altered the thalamic VPL and VPM nuclei regions involved in cerebral somato–nociceptive relay functions. After ODS damage, these were characterized by a spongy aspect developed in the core of tissue degradation where the neuropil damage is located, concomitant to a breach in the blood–brain barrier [67]. While this damage did not result in an immune inflammatory tissue response, it resulted from a cascaded signal from astrocytes to oligodendrocytes [64,65,66], causing a rapid myelin loss as marked by adenomatous polyposis coli (APC) and proteolipid protein (PLP) immunoreactivity that culminated 48 h post-correction of hyponatremia, verified with ultrastructure [62,64,66,68]. Those changes were revealed along with astrogliosis as well as with astroglial clasmatodendrosis (cell body swelling with fragmentation of distal processes) [63,64,65]. In this murine model of ODS, the CNS thalamic regions involved comprised 150–250 µm wide regional disengagements of capillary endothelial tight junctions resulting in the leakage of plasma components, surrounded by overloaded microglial cells amongst the lesioned degradations of the neuropil [62,64,66,67,68]. The lesioned ODS zone epicenters were surrounded by a degraded gradient tissue that contained necrotic and less damaged nerve cell bodies with demyelinated extensions. There, in the outskirts of the lesions, nerve cell bodies and oligodendrocytes revealed irreversible damage but morphologic trends of restoration [66,67,68].

In the present report, we describe peculiar findings concerning ADP-ribosylation factor-like protein 13B- (or ARL13B)-labelled structures, first found in a few resilient oligodendrocytes [68] that are also located in and from the axon hillock regions of resilient nerve cell bodies as the axon initial segments (AIS) appeared in the ODS outskirts of the lesioned regions and extended distally. Even though thalamic neurons seem not to degenerate inside ODS demyelinating lesions, several reports from our group using TEM analysis pointed out signs of cellular adaptation at the level of their neuronal cell bodies. Among them, neuron cell bodies early emerged from the ODS challenge with deeply indented nuclei owing nucleolus translational activation, huge amounts of polysomes along with secretory-like activities [66,68,69]. Another main fine structure revealed at the axonal hillock was a primary cilium appendage that corresponded with contrasted rod-shaped structures or shaft larger growths in the ODS surrounding regions after chronic hyponatremia, 12 h and 48 h time lapses following rapid reinstatement of sodium balance. These accumulated ribonucleoproteins progressed into organized cytoskeletal components along the AIS segments issued from the axon hillock region. These axonal reorganizations post-ODS could reflect the latent potential for this murine model as with the favorable outcomes in the human ODS cases, where nerve cell repairs could explain the occurrence of some restoration of thalamic somato–sensory functions.

## 2. Results

### 2.1. Light Microscopy Aspects of ODS Thalamus

The blood of mice included in the protocol was assayed for serum Na^+^ (SNa) along different conditions of experimental ODS (Figure 1 and Table 1); in particular, prior to correction of hyponatremia and at 24-h post-correction, which delta (∆SNa) usually predicts the occurrence of CNS demyelination. The average of ∆SNa at 24-h post-correction in our cohort of ODS mice was 24 mEq/L.

Figure 2 displays thalamus changes as a set of histology parasagittal sections of NN, HN, ODS 12 h and ODS 48 h murine brains, stained with Eriochrome C. Adjacent sections were used for MBP and ARL13b immunochemistry studies. As shown in Figure 2, of the parasagittal sample brain sections from each animal group, it is only in the ODS48h-treated mice where the thalamus reveals a clear, approximately 1–1.2 mm diameter-wide pale-stained posteroventral region. There, a regional thalamic myelinolysis is revealed by the loss in the Eriochrome staining and MBP immunoreactivity (asterisk in Figure 2) while, in the other treatment groups, no obvious difference in overall myelin staining intensities was observed. In the ODS48h, thalamic ventral posterolateral (VPL) and ventral posteromedial (VPM) relay nuclei to contain the worst demyelination region, and a typical Eriochrome hue background is recovered within a 30–50 µm narrow distance away from the outskirts of the 200–250 µm damaged core zone, indicating there is a somewhat blurred degradation limit for the damaged brain region, revealed with a distinctive paler contrast after chronic hyponatremia compared with the other brain treatment groups sampled. 

### 2.2. The ARL13B Labeling and the Resilient ODS Thalamic Nerve Cell Bodies

A prompt survey of microscopic anatomy preparations marked with ARL13B immunolabels, as shown in Figure 3 and Figure 4 reveals with light microscopy the thalamic tissues with an overall yellow–ochre hue from which scattered whole-like, oval to round shapes appear as ‘holes in the fabric’ corresponding to the nerve cell bodies, either isolated or as joined pairs at the inner outskirts and at least 300 µm away from the thalamic ODS demyelinated core zones. There, nerve cell bodies appear decorated by pale to dark brownish protuberances to rod-like shafts, taking the aspect of either straight or bent poles. These structures are prominent due to their contrast, labeled shapes caused by diaminobenzidine deposits, and always appear associated to the scantily contrasted thalamic nerve cell bodies (Figure 4) throughout the fields-of-view examined. From their morphology, one could recognize these rod-shaped neuron appendices as AIS. At first glimpse, these ‘appended’ structures are of diverse length whose roots seem thicker than the distal, tubular elongated part, of equal diameter but of various lengths, according to random sectioning. Preliminary measurements of the appendage lengths and frequency seem to vary according to experimental group and location related to the myelin degradation edge distance and show these neurite shafts as more numerous and elongated distally from the epicenter of the ODS damaged region as well as with a longer time lapse after chronic hyponatremia is rebalanced (Figure 5A,B). Thus, in a sample of LM random field of view, those labeled ARL13B NN, HN, ODS12h and ODS48h thalamic structures are illustrated, and exemplify the marked contrast shown by the AIS structures that appear in the CNS tissues where nucleoli reveal transcripts in ODS48h > ODS12h > HN appearances. Furthermore, as shown previously in the thalamic nuclei, some nerve cell bodies display a twin association as they join by ephaptic contacts [69]. The specificity of ARL13B immunolabeling is verified using technical and biological controls (Figure A1). 

### 2.3. The ARL13B Immunolabels in the Axon Hillock to Extend in the Axon Initial Segments

Other cell types, including those of the neuroglial types and capillary endothelial cells, can be found among the light brownish hue background. Figure 3, Figure 4, and Figure 6A also convinced us about the ARL13B labeled structure when, as shown in Figure 6B,C, toluidine blue stain allowed us to show similar cell extensions of ODS thalamic nerve cell bodies in epoxy 1-µm thick sections, where short but chubby axon hillocks extending into AIS were strongly stained. It is only through using TEM views that equivalent structures were identified and deciphered with ultrastructure, such as those illustrated in Figure 7A–C. In the meantime, both closest to the damaged edge of the chronic hyponatremia demyelinated zone, as shown in Figure 3 and Figure 4, HN and ODS12h as shown in Figure 6A–C, other immunolabeled cone-shaped appendages appeared to issue from the neuroplasm whose outline offered the same intense brown contrast found in Figure 6A. In Figure 6A, the hemalum stained the nucleolus in a blue and purplish hue and the epoxy equivalent structure toluidine blue-stained sections revealed an intense basophilia almost as if each heavy spike issued from the perikarya, close to the nucleus (Figure 6B,C). These features altogether confirm that both LM staining patterns have marked and recognized enriched parts of the axon hillock that formed and extended into AIS. In addition, they were repleted with both freed, polyribosomal and attached endoplasm ribonucleoproteins amongst few mitochondria, but still leaving enough intracellular neuroplasm to gain an overall toluidine blue orthochromaticity (Figure 7A–C). Again, with a fine structure, those ODS12h and ODS48h thalamic nerve cell body features revealed nucleoli components that comprised abundant transcriptions and translations. The same ODS12h thalamic nerve cell bodies contained vestiges of physiological stress that showed as fine injuries that altered the neuroplasm contrast, facing all the nuclear pores, and some endoplasmic reticulum cisterns or parts were seen as if the scratches of nails were left among them (Figure 7B).

### 2.4. ARL13B in ODS12h: LM and TEM Aspects of Neuron Extensions

Reviewing micrographs obtained from past data of ODS along with these observations, the murine CNS thalamic region investigated at the time lapse ODS12h can be exemplified where an approximate surrounded and underlined part corresponded with that part which underwent demyelination, as shown in Figure 8A. There, an ODS core area with wasted neuropils showing among its spongy landscape aspect can be seen. In the immediate outskirts, some neurons can be found with LM (Figure 8B) and parts of them were still recognized with TEM (Figure 8C). As shown, in this latter micrograph, two adjacent ODS core neurons demonstrated their necrotic deterioration due to irreversible injury, as their heavily damaged morphology demonstrated nuclei that underwent chromatolysis, revealed by diluted chromatin or remnants whose nucleoli had vanished leaving vacuolated neuroplasms. There, a loosened endoplasm, and several lysosomal and lipofuscin bodies remained. Additionally, when a satellite oligodendrocyte was still recognized, it was as a shrunk, lytic deteriorated body that bore a compacted necrotic nucleus. Within 100–150 µm, the distance away from the worst core damage of the ODS epicenter made of deteriorated neuropils, some of the nerve cell bodies displayed resilience by showing remaining funnel-shaped to elongated shaft-shaped projections issued from their perikaryal zones and, in the best findings we were able to observe, extensions ranging from 2.5 to 15 µm in length (Figure 6, Figure 8, Figure 9, Figure 10, Figure 11, Figure 12, Figure 13 and Figure 14), also evidenced by the ARL13B labeling (Figure 3, Figure 4, Figure 6A, Figure 13A and Figure 14A). At first, micrographs collected with TEM showed numerous granules in these extension that corresponded with the high ribonucleoprotein content associated or not with elongated cisterns of endoplasmic reticulum reaching the core of these axon extensions and, while reaching the funnel constriction segment, the heavily proteinaceous content revealed numerous paraxial and parallel neurotubules (Figure 8D,E). An enlarged view in Figure 9 seems to carry the fine structure aspect of this growth extension of the axon hillock, known as the axon initial segment (AIS) where innumerable proteinaceous components, including neurotubules and associated cytoskeletal macromolecules, appear arranged in periodic rows along and perpendicular to or encircling the cylindric AIS shape and neurolemma can be viewed. The neuroplasm itself, with an enlarged view (Figure 9 insert), can show an underlined parallel concentration of particulate proteins that could be called a sub-neurolemmal structure, further suggesting the unique and crowded peculiar architectural cytoskeletal components necessitated to construct and grow this special segment of the nerve cell body extensions.

### 2.5. ARL13B Label in ODS48h: LM and TEM: A Primary Cilium Emergence

The thalamic neurons located adjacent but at less than 100 µm distal from the necrotic core ODS, as seen in Figure 8A–E, and more clearly in Figure 9, can be recognized from their shape, using LM, as a large pale nucleus with indent(s) that reveal an evident stained nucleolus and, with TEM, the same highly contrasted, large nucleolus shows large amounts of accumulated granular transcripts (as granular component) accompanying the dense and fine fibrillar components of the chromatin distributed throughout the very active nucleoplasm (Figure 9A, Figure 10A, Figure 11A and Figure A2). At first glimpse, the adjacent neuropil and the satellite oligodendrocytes appear typically located attached to neurons as satellites but, with TEM scrutiny, these were mostly damaged by ODS cell stress and are surrounded by large neuropil intercellular spaces throughout and possess other cells’ and myelin remnants, implicated after leakage of blood fluids, plasma with serum, contributed via the small regional necrotic cells to the archetypal term of ‘liquefaction necrosis’. However, within this ODS damaged zone outskirts, among some degraded and corpses of neuropil, resilient neurons found in the outskirt region were marked by large euchromatic nuclei and active nucleolus; they showed numerous ribonucleoproteins but also reveal through their fine structure that they possess delicate but clear-cut primary cilia (Figure 10A,B). Each cilium length ranged from 4.5 to 6.0 µm long and width from 0.2 to 0.25 µm that is filled with a core of granular cytoplasm and microtubules. The example illustrated shows its transition zone cell attachment consisting of a narrow transition segment or ‘neck’ of about 0.15 µm in width, that is tied by basal fibrillar materials, and forms a ciliary pocket or cove-like space of the plasma membrane. Within the neuroplasm, obscured by numerous ribosomes, a microtubule fascicle and other subcellular filaments can be barely seen originating from the adjacent perikaryal region and reaching the basal side of the primary cilium whose random sections constitute a sort of ‘hub’ region, probably endowed by a centriolar piece structure in some other ultrathin section. In Figure 10A–C, adjacent to the primary cilium, one can note filopodia-like extensions, as these could also be relevant to this rejuvenated change associated with cell resilience (Figure 10C). Other neurons of the same region (Figure 8D,E), at low magnification, bear an axon hillock that becomes like a funnel, similar to the neurons, and more distally located away from the core ODS damages, as shown in Figure 11, Figure 12, Figure 13, Figure 14 and Figure A2, displaying axon hillocks and an axon initial segment without any ciliation. Underlined by ocher yellow (Figure 12A) and enlarged in Figure 12B, the Golgi apparatus saccules characteristically erupt in numerous small vesicles (50–70 nm diam) that accompany ribonucleoprotein particles (mainly RNAs), revealing swirled ribbons of polysomes in places. Vesicles and particles appear to redistribute into the AIS, showing as an initial squeezed cylindric shape aligning the entered neurotubules into a paraxial to parallel orientation congested by one or more long, twisted mitochondrion. Both mitochondria profiles and inner neurolemma leaflet can be decorated by small round to elongated vesicles while the outer leaflet has typically no myelin. However, some synaptic contacts are noted amongst end-feet profiles of astrocytes, recognized by their highly contrasted granules of glycogen content (Figure 12A,B).

### 2.6. ARL13B Label in LM and TEM Thalamic AIS Growth of ODS48h Neurons

From the axon hillocks, with ARL13B labeled, the AIS also shows ARL13B labeling. The overall rod structure shapes viewed with LM (Figure 13A,B and Figure 14A,B) can be confirmed by TEM as containing fascicles of neurotubules fashioned into a type of furrow where mitochondria can align, revealing their snaked profiles (Figure 13C and Figure 14C–F). All the AIS depict synaptic contacts and only the astrocyte end-feet amongst a neuropil still reveal discrete or prominent damages (stars) with intercellular spaces resulted from ODS-stressed tissues.

### 2.7. ARL13B Protein Expression in Thalamus Homogenates

Data from excised punches of the contralateral halves of the murine brains of the same ODS experiment groups found no significant differences between the ARL13B protein expressed (Figure 15 and Figure A3). However, the ODS core zone of the thalamus nuclei revealed neuron tissue necrosis, where no or low level of ARL13B protein would show in assays. In fact, the necrotic zone encompasses only a narrow epicenter zone of the thalamus region within the ODS regional damage that ranges from 150 to 350 µm [62,64], and the excised punches were approximately 500 to 600 µm. The surrounding demyelination damage presents a sort of gradient of neuropil damage where resilient neurons remain within 200 µm distance, and those possess either a primary cilium or exocyst emerged out of axon hillocks where some ARL13B protein are expressed, as shown with immunolabels in all the HN and ODS12h and ODS48h samples. There, ultrastructural findings described and discussed in the above paragraphs confirm the data found with the blots.

## 3. Discussion

### 3.1. Clinical Considerations

Following the human brain autopsies and the histopathology findings of Adams and others (1,2), symmetrical and bilateral demyelination in the pons, called ‘central pontine myelinolysis’ (CPM), revealed that nerve cell bodies were not altered, and the associated changed tissues showed no inflammatory reaction [1,2,3,4,5,6,7,8]. Following these earliest reports, other cases with patients showing demyelination damages were in other CNS regions, such as the cortex, the thalamus, the basal ganglion, the cerebellum, and the hippocampus. Thus, these locations away from the pons were each called ‘extrapontine myelinolysis’ [1,2,3,4,5,6,7,8,9,10,11,12,13,14,15,16,17,18,19,20,21,22,23,24,25,26,27,28,29,30,31,32,33,34,35,36,37,38,39,40,41,42,43,44,45,46,47,48]. In the meantime, the etiologic link between demyelination and perturbations lasting more than 48h in the homeostatic level of [Na^+^], formerly named central pontine myelinolysis, a chronic decrease of less or above homeostatic balance than 1 mEq/L/h made hyponatremia occur as the blood sodium level went below 135 mEq/L, has been found through a retrospective review of medical records related to ODS cases between 1960 and 2018 in the United States National Library. These were identified through routine hospital blood analyses as hormonal dysregulations, e.g., [45,46,47,48,49,50,70,71,72,73,74,75,76,77,78,79,80,81,82,83,84,85,86,87,88,89,90,91] that could be associated with post-surgery [85,86,87,88,89,90,91,92]. Except for one report, where CPM was stated to be without hyponatremia [70], surveys of hospital cases between 1999 and 2018 from the Massachusetts General and Brigham and Women’s Hospitals, Boston and, ”using International Classification of Diseases–9th edition codes and a text-based search for central pontine myelinolysis, extrapontine myelinolysis, and osmotic demyelination syndrome” [5,6], many other clinical examples of chronic hyponatremia with ODS were found in obstetric care units or in pediatric care units [93,94,95,96,97,98,99,100,101,102,103,104,105,106,107,108,109,110]. In the neonatal and pediatric units, about 2.5% of emergency care ODS patients can die from it [6,45,80,97,98,99,100,101,102,103,104,105,106,107,108,109,110,111,112]. Another survey of emergency admissions to hospital showed hyponatremia can occur in 15–20% of critically ill patients [113,114]. The same proportion can be reported for the confined, aged population [115,116,117,118,119,120], where hyponatremia could cause or result in motility-related disabilities (such health incidents could be preventable) because deficits in homeostatic sodium gradient along with mental disorders can cause patient’s confusion, disorientation, paresis—including fracture susceptibility—[85,114,116,120,121,122,123]. In other cases, deafness, memory loss to seizure, unresponsiveness and eventually coma have occurred, e.g., [21,28,50,124]. Additional case examples of ODS can be found in patients affected by other clinical situations where sodium out-of-balance events were associated with or resulted in kidney function disorders [45,46,47,88,89,90,97,125,126,127], hepatic disorders [91,128] and transplantation [128,129,130,131], thyroid Grave’s disease and diabetes [132,133], cardiology care, e.g., [134,135,136,137,138,139], drug interactions [140], and in those patients with autoimmune infections [141]. Finally, specific heat-exhausting exercises where dehydration could create prolonged sodium imbalance and hyponatremia [142,143,144,145,146,147,148,149,150,151,152]. In total, beyond inborn errors in myelin formation [153,154], it is probable that the patients with ODS-associated defects are underdiagnosed, due to lack of systematic brain imaging and/or occurrence of subclinical ODS forms.

### 3.2. The Translational Aspects of the Emergence of a Primary Cilium among Neurons Outside the ODS Epicenter

The LM yellow–ochre hue aspects of the ARL13B immunolabeled thalamic tissues appear as if covered with an overall from which scattered hole-like, oval to round shapes appear as ‘holes in the fabric’, where the nerve cell bodies, either isolated or as joined pairs or triplets exhibit poor contrast, as shown before with resilient nerve cells [66]; meanwhile, they maintain low resistance contact junctions with each other, as noted in Figure 12A [69]. However, the nuclei, poorly stained, disclose orthochromasy where contrasted nucleoli in pale blue staining according to hemalum reveal accumulated RNA transcripts with a chromatin DNA-stained dark blue hue [155,156]. Similarly, the semi-thin epoxy sections, prepared to choose zones to be ultra-cut, stained by toluidine blue, allow us to verify the abundant transcripts located in an appendix-like zonation with profused ribonucleoproteins [157,158]. Here, the axon hillock extends into the ARL13B labeled segment that is similarly richly endowed by transcripts, detected where the primary cilium and a future AIS grew after ODS, away from the demyelinated ODS epicenter, in the outskirts of the demyelinated thalamic zone. The resilient neurons show a perikaryal zone where most of the cells typical organelles and inclusions are located, including the Nissl bodies, contained cytoskeletal components among sparse polyribonucleoproteins associated or not with the endoplasmic reticulum along with mitochondria and apparently haphazardly oriented neurotubules (also known as microtubules) that are ultimately associated with the changing funnel shape of the cell extension that then possess these neurotubules, associated with neurofilaments and free ribosomes [159]. These neurocytologic components can also be viewed in our resilient neurons illustrated in Figure 7, Figure 8E, Figure 10, Figure 11, and Figure A2 where AIS is forming, as further explained in [159], with the earliest ultrastructural publications. There, according to the ODS lapse of time views, the typical perikaryal neuroplasm showed amassed polyribonucleoproteins (polysomes) within a distal region that yielded a sort of exocyst extended into a comma-shaped and in distal ODS into more or less long cylindrical extensions marked by an ARL13B label. It is after scrutiny of the ultrathin sections of these resilient neurons that a few of these sections (1 out of 15) revealed a display of a primary cilium (Figure 10A,B). These unique observations among ODS CNS damages confirmed those ARL13B labelings, similarly to those revealed for some oligodendrocytes within the same murine ODS damage region [68]. The occurrence of such neuronal primary cilium outgrowth was not investigated elsewhere in the CNS; one of the reasons is that, unlike the thalamic nuclei, the centropontine regions or subcortical regions are inconsistently affected by demyelinating lesions in the model of ODS in mice [62].

This primary cilium cell appendage was observed and described a long time ago [160], including with high morphologic resolution by electron microscope tomography [161]; its physiologic sensing functions have only been recently investigated in numerous normal and pathology cells and tissues as found in all embryonically derived epithelial cells from in vitro and in vivo studies, where this appendage was found to be a mechanical, flow sensing device but also as osmosensor [162,163,164,165]. Its dysgenesis and mutations of the cilium-associated structure and membrane signaling proteins favor insensitivity and growth anomalies such as polycystic kidney disease (PKD), Bardet–Biedl syndrome (BBS) and/or other cranio-facial dysgenesis as a modulator function of tissue’s differentiation [166,167,168,169,170]. In the CNS, the primary cilium appeared crucial in adequate morphogenesis as it was involved in correct neurogenesis in cortical development as well as for axon growth [166,167,168,170,171,172,173,174,175,176,177,178,179,180,181].

Accordingly, the primary cilium phenotype in mature CNS can indicate some revitalized nervous structures and be a marked precursor for other noted repairs [66,68] that confirmed a complex cascade of transduction signals triggered by morphogen growth factors or electric stimuli. As an example, secreted smoothened hedgehog (Shh), a protein signaling bound patch (Ptc) at the cell surface, relieved inhibition of a transmembrane protein, smoothened (Smo), that can further conduct a cascade of transduction activations in the arf family genes that are included the ARL13B expression and are labelled as such in our CNS tissues [68,176,177,178,179]. Ultimately, these emerged cilium and associated filopodia (Figure 11C) structures verified that some restored axon extension would progress as an axon hillock region into a rebuilt AIS and the distal axon segments. Meanwhile, as a side note, it emerged out of these resilient nerve cell bodies filopodia that accompany the primary cilium signal sensor appendage that other mechanosensors should become ablated by endocytosis while further growth and differentiation occurred [180,181] into matured circuitry, involving the AIS and the distal axon interconnections [182,183,184] as influenced by soma location and morphology [185].

### 3.3. Out of ODS: From Primary Cilium toward Axonal Maturation

#### 3.3.1. The Axon Hillock

The formed axon hillock would receive inputs from other distant, preserved neurons as axo–axonic synaptic contacts have been noted in the same illustrations quoted above, and would influence their functionality through signal transductions of diverse renewed, associated ion voltage-gated channels [159,160,161,162,163,167]. As it matured, a sort of barrier appeared between the axon hillock and AIS because internal constituents appeared to restrict some macromolecule passageways [180,185,186,187,188,189,190] but provided the axolemma with adequate insertion of suitable ion channels [164]. This applied in particular to those involved in initiation [159,165] and contributed to the adequate action potential [161], as well as of other surface receptors [166], including those axo-axonic synapses [162], also seen in Figure 8D, Figure 9, Figure 11A,B and Figure 13D.

#### 3.3.2. The Axon Initial Segment (AIS) and ARL13B Label Significance

A cylindrical extension of the axon hillock morphology of the same diameter and of variable length according to its CNS location, development stage and maturity as well as species [159] has been named the ”axon initial segment” or AIS. The AIS fine morphology was first reported in [159,191,192] and many other publications, including recent recurring surveys, due to the progress in an understanding of its molecular components that has ensued [193,194,195,196,197,198,199,200,201,202]. The main characteristic is certainly that its axolemma, free from myelin—as not yet enfolded by the oligodendrocyte extensions—appeared usually as a tubular, shaft-shape that distally extended into the axon proper, the longest extension of most neurons [159,195,203]. The changes in length and maturation of the axon hillock combined with the AIS CNS location seemed to adjust the ion-voltage channel populations and distribution as they can influence excitability [197,204,205]. AIS of the ODS12h and, more evidently, those of HN, ODS12h, and ODS48h revealed morphologic features classically found by other morphologists [159] to be without Nissl bodies; however, here basophilic components were not excluded because, during the post ODS lapses of time investigated after rebalancing osmolality, AIS presented high electron density of ribonucleoproteins that verified abundant translational activities [66], probably like those noted during development. These abundant transcripts, spilling out of the reactivated nucleolus (Figure A2) as in [66], were revealed specifically as many LM molecular markers with high resolution microscopy [206] and biochemical analyses [200]. These ribosomes and polyribosome population dispersed among other organelles, including the mitochondria that showed even in AIS under restoration (Figure 8E, Figure 12A,B, Figure 13C and Figure 14E,F). These observations concurred with those of others because mitochondria were also found with ion beam/scanning electron microscopy (FIB/SEM) in this neurite segment that underwent repairs [206,207,208]. Out of the aforementioned surveys [193,194,195,196,197,198,199,200,201,202], the AIS complex of macromolecular assembly and architecture was progressively deciphered by a few in vivo, organoid, and many in vitro studies from the earliest soma protrusion to form an axon [209,210]. With a survey that quoted more than 45 macromolecular glycoproteins and proteins whose location in the AIS have been individually located, one can summarize the AIS as subdivided into three cytologic concentric layers:

(a) The membrane or axolemma, in contact with the neuropil, and a sleeve of about 100 nm thick neuroplasm (Figure 9, Figure 11, Figure 12 and Figure 13) is different from the AIS core extended in the distal axon. There, the major AIS scaffolding protein ankyrin G (AnkG) and *β*IV-spectrin interacted with many other proteins of the axolemma, including voltage-gated ion channel sodium ((Nav) that include Nav 1.6, Nav 1.2, Nav 1.1), potassium K(+) (Kv1.1, Kv1.2, Kv1.4, and Kvbeta2) glycoproteins which bestow unique electrical properties of the AIS, whose axolemmal distribution depended on development stage and CNS location [188,193,195,211]. Those channels’ diversity and distribution density remained somewhat unclear [171,182,211,212,213], but are important because the axolemma of the initial segment is where the action potential originated for initiation of neural functioning and regulation. It is similar with Ca^2+^ subunits and adhesion receptors (e.g., L1 family neurofascin [NF-186]) whose distributions and co-localization also appeared within the phospholipids of axolemma. Those channels interact with the sub-membranous molecular complex layer (see the next paragraph) composed of membrane scaffolds, i.e., cell adhesion molecules and cytoskeletal proteins exhibited a similar dense granular layer as submembranous (Figure 9, insert, and Figure 11B); also seen in the specialized membrane cytoskeleton of the nodes of Ranvier [159,214,215]. Possibly, the heterogeneous vesicles, born out of the RER and Golgi sorting, such as those shown in Figure 10C,D, can confirm those transcripts, translated at the endoplasm membranes and Golgi to be dispatched at their proper docking axolemma and axoplasm sites. Throughout and along AIS external axolemmal leaflet, adhesion, and recognition molecules such as membrane-anchored proteins ADAM22 (disintegrin and metalloprotease), contactin-associated protein-like 2 (Caspr2), transient axonal glycoprotein-1 (Tag1), and the postsynaptic density-93 (PSD-93) can be found.

(b) A sub axolemmal axoplasm layer, 70–100 nm thick, whose cytoskeleton contained AnkG, a layer of α2- and β4-spectrins tetramers, attached to the neurolemma with AnkG, twisted rope-rings of actin crosslinked with α-adducin, and revealed a periodicity of 120 nm. There, phosphorylated myosin II light chain (pMLC), tropomyosin 3.1 (Tpm3.1) and a proteasome (ECm29) were located. However, this cytoskeleton remained with some unclear function(s).

(c) The inner axoplasm is constituted of the same adhesive protein AnkG, that was suspended in the inner central neuroplasm, issued from the axon hillock centriole bodies, neurotubules (also known as microtubules), at first apparently haphazardly distributed in the axon hillock, became organized at the AIS level into a funnel shape, also crosslinked by AnkG tails, as an inner tubular cylinder that prolonged into the distal axon. These neurotubules and associated proteins (also known as MAPs), such as tau, end-binding proteins (EB1/EB2) [196,198,200] crosslinked by the tripartite motif 46 (TRIM46)-containing proteins [216] and organized by activator p35 Cdk5 and p35 [217,218,219] constructed a set of parallel, cylindric-shaped bundles that appeared from LM with high resolution like a “meshed guard fence’” framed by actin rings suspended in a centrally located bundling where migrating or transiting compounds, dispatched as issued from the Golgi apparatus, found in somata would have to obtain clearance to pass within such a ”tunnel” towards the AIS and distal neuroplasm where these transports occur via kinesin interactions [189,220]. Additionally, between the AIS neurolemma and the spectrin outer lining, the cytosol or neuroplasm includes proteins needed for axon proper guidance, such as the large multidomain MICAL protein and a garland network series of septins 5, 7, and 11 that demarcate a zone enriched by myosin II and actin-related protein Arp2/3 [219,221]. However, neurofilaments (10 nm diam. or intermediate filaments) have been found in the typical, squid neuron cell somata [222] while, in mammals and humans, they seemed to be normally also confined to dendrites and synaptic structures [223] and were in the AIS of neuropathologic cases [224]. However, these fibrillar components have been parts of neurotubule fascicles and part of the electron dense finely granular material undercoating of the axolemma along with dispersed ribosomes with early fine structure descriptions of neurons, albeit without identifying the marker (as reported in [159]). In this report, some enlarged views appeared as linkers or spacers with periodic intervals ranging from 28 to 40 nm, and many of these spacers showed dark dots sprinkled throughout with a similar electron density as ribonucleoproteins whose resolution was a scaffold-like fine structure associated with microtubules. However, the lack of molecular markers for fine structural identification made us unable to claim them in the found AIS that grew (Figure 8E). Similarly, as shown, hints of some periodic alignment of polar proteins among the ribosomes amassed suggested AIS components under formation, associated with the axon hillock extension and were suggestive about the complex-built ups and interactions occurring therein as reported elsewhere, e.g., [193,194,195,196,197,198,199,200,201,202,225].

#### 3.3.3. The AIS and ARL13B Label Marker Significance

ARL13B protein is also named ADP-ribosylation factor2-like 1 protein known in the cells of most living plants, animals [226,227], and humans [228,229]. Some of the many human defects involving the genetic and cranio-facial anomalies were reviewed in [166,167,168,169,170]. The gene coded ARL13B is located on chromosome 16 in the C57BL/6J mouse strain used in this study. It is part of the Arf (ADP-ribosylation factor) superfamily that in mammals includes at least 22 members, including the six Arf family members that were found in very early eukaryotes. As also indicated in [226,227,228,229], this protein is localized in the cilia, plays a role in cilia formation and in the maintenance of cilia [228]. In human tissues, ARL13B protein data collected in [228] showed a confidence level of ARL13B protein subcellular abundance that included the sequence: cytoplasm > cytoskeleton > plasma membrane > Golgi apparatus. This sequence can confirm our observations associated not only with the sustenance and formation of the primary cilium components [229]. The primitive developmental gene smoothened Shh led a cascade family of ARF transcripts and other signal transductions associated with renewed synthetic activities assisted by some small GTPases [230,231,232,233,234,235,236,237,238,239,240,241,242,243] that contributed to the homeostasis, growth and restoration of the axon hillock extended by its AIS and distal axon parts. All sorted and dispatched vesicles by the Golgi apparatus can transfer to the proper sites of the AIS axolemma and content assembly that regulated the phospholipid composition of the ciliary membrane, including signaling molecules [176,177,178,179,230,231,232]. About these distributions, the Golgi hetero-contrasted vesicles we viewed in the adjacent axon hillock could represent a distribution of some of the glycosylated phospholipids, voltage-gated channels, receptors, and adhesion molecules towards the axolemma that may be detected in ODS12h and ODS48h (Figure 11C). Possibly, small GTPases, especially those linked with the Rho reactome, would manage the energetic necessities implicated in this interweaved trafficking and placement of many of the component molecules of the AIS building, not only those neurotubules and MAPs architecture, but also those axolemmal receptors and channels as well as the elements of the sub-axolemmal cytoskeleton and cisterns altogether and could reach variable length [225,226,227,228,229,230,231,232,233,234,235,236,237,241,242,243,244,245,246,247]. In particular, the thalamic neurons can be highly dependent in maintaining Na^+^ and K^+^ channels [248,249] and placed accurately to avoid pathology [249]. Meanwhile, those active transductions would have dealt with axon guidance and neuronal protrusion [240,244,245,250], whose construction was properly orchestrated with the microtubule- and molecule-associated (MAPS), to build the AIS baton [177,178,179,226,227,228,229,230,231,232,233]. Hence, the ARL13B marker or labeling appeared at first as a perikaryal extension of the axon hillock as shown in Figure 7 in HN condition and, later, in the ODS12h samples and both with toluidine blue. These regions showed a comma-shaped intensely basophilic segment, without any metachromatic characteristics. Verified with TEM, these intensely basophilic regions were seen enriched in polyribosomes and a few elongated cisterns of endoplasm, suggestive of a peculiar and huge Nissl-like body where the ARL13B protein would participate in the built-up components that extended the axon hillock into the axon initial segment. These accumulated transcripts would show some similarity, as in the growth cones [243,244,245,246,247,250].

### 3.4. From Mice to Human: Translational Considerations or Can the CNS Recover from ODS Damages?

The murine investigation model of ODS that we developed [62,64,65] would suggest that after regional CNS damages, the altered thalamic regions can recover some of the relay thalamus functions following tissue repairs according to their location from the damaged epicenter, that has undergone necrosis where regional demyelination accompanied a classic ‘liquefaction necrosis’ after progressive rebalanced osmolality [62,64,65,66,67,68,69]. Accordingly, the investigated time lapses after reinstatement of osmolarity indicated some neural and oligodendrocyte resilience and reorganizations post-ODS. Thus, longer time lapses could be investigated to further understand the progress in the CNS tissue resolution e.g., glial scarring, recruitment of oligodendrocyte progenitors and potential delayed remyelination [251,252,253,254,255]. In the present animal model, ODS 12h and 48h were limitation lag times chosen for the investigation, conditional on the amount of research support granted and its implementation, based on previous studies effected in other mammal species [55,56,57].

A similar question remained about astrocytes that appeared at first to ‘alarm’ or be keyed to trigger the oligodendrocyte changes with demyelination damages while the blood–brain barrier was breached in ODS [57,65]. The restoration of oligodendrocyte functions started after ODS12h [68] but that of astrocytes lagged. Could this be due to astrocytes’ differences in damages (i.e., protoplasmic vs. fibrillar) and/or age like for patients [256,257,258]? Considering the latest information about demyelinating defects and some remyelinating potentials illustrated here one still remain to know whether with delayed time after ODS damages, a sort of modus vivendi between astrocytes, oligodendrocytes, and neurons signals in making remyelination to return ODS thalamus or other CNS-damaged regions completely able to retrieve standard structures and functions. Possibly, similar repairs occurred in human ODS where most patients appeared to have a favorable clinical prognosis in the short-term post-hospitalization even though, retrospectively, there were and are still concerns for a range of 10 to 25% fatal outcomes [22,28,45,107,108,109,110,111,112,113,114,115,116,117,118,119,120,123,259,260], or other neurologic defects [112,261], especially due other underlying disease states [45,259,260,261,262].

## 4. Materials and Methods

### 4.1. The Animals

Male C57bl/6J mice, aged from 3 to 4 months were kept in the University Animal Facility. According to the ODS protocol, animal experiments were conducted in compliance with the European Communities Council Directives for Animal Experiment (2010/63/EU, 86/609/EEC and 87–848/EEC), approved by the Animal Ethics Committee of University of Namur (ethic project number UN 14–210).

### 4.2. The Murine ODS Protocol

ODS induction was based on the correction of chronic hyponatremia, according to an adapted protocol from [55,57], as also described in [62,64,68]: an osmotic minipump (Model 1004; Alzet, Cupertino, CA, USA) was filled with desmopressin acetate (2 µg/mL; Minirin, Ferring, Saint-Prex, Switzerland) and inserted subcutaneously under anesthesia into the back of animals on day 0. Standard pellets and water were switched to a low-sodium liquid diet (AIN76A; MP Biomedicals, Santa Ana, CA, USA), given ad libitum for the whole duration of hyponatremia. At day 4, hyponatremia level and serum sodium were increased back to normonatremia using a single intraperitoneal injection of NaCl 1M (1.5 mL/100 g body weight). Serum [Na^+^] level was measured using Spotchem EL SE-1520 electrolyte analyzer (Arkray, Kyoto, Japan) according to the manufacturer’s instructions. Minipumps were left in the mice until the end of experiments. Unless otherwise specified, any procedure involving anesthesia was performed using intraperitoneal injection of a cocktail of ketamine 100 mg/kg and xylazine 5 mg/kg.

### 4.3. ODS Experiment Groups

This ODS investigation encompassed thirty-nine mice, subdivided into four groups. Group 1 were normonatremic or sham mice (NN; n = 13) sacrificed at day 0; Group 2 were hyponatremic mice (HN; n = 11) sacrificed 4 days after the induction of hyponatremia (day 0 + 4-day treatment period named ‘chronic hyponatremia’, as described in the ODS protocol [62,64,66,68]. Groups 3 and Group 4 were mice that underwent the 4-day ‘chronic hyponatremia’ abruptly provided with normonatremia both labeled with the acronym ‘ODS’. Group 3 included mice sacrificed 12 h after a fast restoration of normal natremia, thus named the ODS 12h group (ODS12h; n = 6) while Group 4 mice encompassed mice sacrificed 48 h post fast osmotic correction, hence named ODS48 h (n = 9) (Figure 1).

### 4.4. Light Microscopy (LM): Morphology and Immuno-Histochemistry

Under anesthesia, all the mice were exsanguinated and perfused transcardially with warm NaCl 0.9% followed by phosphate-buffered 4% paraformaldehyde (PFA). Brains were removed, divided into two hemispheres, and post fixed overnight in the same PFA fixative solution. For histology, brains were then dehydrated, paraffin-embedded and sectioned into 6 µm thick microscopic preparations that were alternatively used for general topographic observation with Eriochrome C for myelin stain as in [62,64] or for other immunolabeling. The paraffin sections were dewaxed, rehydrated and heat-induced antigen retrieval was performed in citrate buffer pH 6 at 100 °C for 10 min. Endogenous peroxidase was quenched using 3% H_2_O_2_ in methanol for 10 min. Non-specific binding was blocked using 5% horse or goat serum diluted in Tris-buffered saline (TBS) for 15 min; the preparations were then incubated overnight at 4 °C with primary antibodies against myelin protein MBP (1:500; Abcam, ab40390) or the marker of ciliogenesis Arl13b (1:1000; Proteintech, Fisher Sc. 11711-1-AP) diluted in TBS containing 1% normal serum. Further, sections were incubated with a biotinylated secondary antibody (1:100, Vectastain; Vector Laboratories, Burlingame, CA, USA) for 1hr at room temperature and contrasted peroxidase-bound streptavidin (1:100; Vectastain) for 45 min. The detection of immunolabeled sites was revealed using diaminobenzidine (Dako, Glostrup, Denmark). Immunolabeled sections were finally counterstained with hemalum, dehydrated and mounted in DPX and observed with an Olympus BX63 microscope (Olympus, Tokyo, Japan) equipped with Hamamatsu Orca-ER camera; images were acquired and analyzed with Cell Sens software, in a non-blinded fashion.

### 4.5. Transmission Electron Microscopy (TEM)

The fine structure investigation complemented those previously performed with neurophysiology, histology, and immunohistochemistry where four groups were used: Group 1 was normonatremic mice (NN; n = 2) sacrificed at day 0; Group 2 was hyponatremic mice (HN; n = 2) sacrificed 4 days after the induction of hyponatremia (day 0 + 4-day treatment period) of ‘chronic hyponatremia’ as described in the ODS protocol. Groups 3 and 4 were mice which underwent the 4-day ‘chronic hyponatremia’ abruptly provided with normonatremia as both ODS Groups, i.e., Group 3 included mice sacrificed 12 h after this fast restoration of normal natremia, thus named ODS 12h group (ODS12h; n = 3), while Group 4 mice encompassed mice sacrificed 48 h post osmotic correction hence named ODS48h (n = 3). Under anesthesia, mice were perfused transcardially with a solution of PFA 2% and glutaraldehyde 2% in 0.1 M phosphate buffer (pH 7.4). Selected brain regions were harvested and post-fixed in glutaraldehyde 4% for 2 h. Thalamus ventral posterolateral (VPL) and ventral posteromedial (VPM) thalamic regions were harvested using a neurological punch of 0.69 mm of internal diameter (#18036-19; Fine Science Tools, Heidelberg, Germany) and VPM and VPL nuclei were sampled (lateral plans 1.0 to 2.0 mm from interhemispheric fissure) [62,64,66,68] according to the mouse brain atlas of Franklin and Paxinos [263]. Samples were washed in Millonig buffer containing 0.5% sucrose for 24 h and were then post-fixed in OsO4 2%, dehydrated and finally embedded in epoxy resin. Semi-thin sections were stained with toluidine blue to choose selected regions of interest for fine structure analyses. Ultrathin grey (interference color) sections (ranging from 40 to 65 nm) of these samples, obtained with a diamond knife, were collected on 200 and 300 mesh nickel grids (Micro to Nano, Haarlem, The Netherlands) and contrasted with uranyl acetate and lead citrate to be observed in a Philips Tecnai 10 electron microscope, at an accelerating voltage of 60–80 kV, equipped with a digitized Olympus ITEM platform MegaView G2 image analysis.

### 4.6. Western Blot

Proteins were extracted from micro-dissected thalamus samples using a lysis buffer containing 0.5M Tris-HCl (pH 6.8), glycerol, 10% sodium dodecyl sulfate, and 200 mM dithiothreitol as in [62,64,68]. Samples were boiled at 100 °C for 5 min and centrifuged at 13,000 rpm for 3 min at 4 °C. Protein concentration was measured using the Pierce protein assay kit (Thermo Fisher, Bleiswijk, The Netherlands). Ten µg of proteins was loaded on a 10% polyacrylamide gel, separated via SDS-PAGE (120 V for 1 h 30) and transferred to a PVDF membrane (100 V for 30 min). Membranes were incubated with blocking buffer (5% NFDM-TBS-Tween 0.1% or 2% BSA-TBS-Tween 0.1%) for 30 min, followed by an overnight incubation at 4 °C with primary antibodies, diluted in corresponding blocking buffer. Primary antibodies used were anti-ARL13b antibody (Protein Tech; # 1771-1-AP, 5% NFDM) and mouse anti-GAPDH antibody (Sigma; #G8795, 1/10,000, 5%NFDM). Membranes were then rinsed and incubated with an anti-rabbit, or an anti-mouse HRP-linked secondary antibody (Cell Signaling; #7074S or #7076S) diluted 1:1000 in the same blocking buffer. Signal was revealed with a chemiluminescent substrate (BM chemiluminescence blotting substrate (POD); Roche Diagnostics, Mannheim, DE) and captured with an Image Quant LAS 4000 mini (GE Healthcare, Diegem, Belgium).

### 4.7. Statistics

All results were expressed as mean values ± standard error of the mean (SEM). For the experiments involving the comparison of more than two conditions, the statistical significance was assessed using one-way ANOVA followed by Dunn’s comparison test. The level of significance was set at *p* < 0.05. All statistical analyses were performed using GraphPad Prism (GraphPad Software 9, La Jolla, CA, USA).

## Figures and Tables

**Figure 1 ijms-24-16448-f001:**
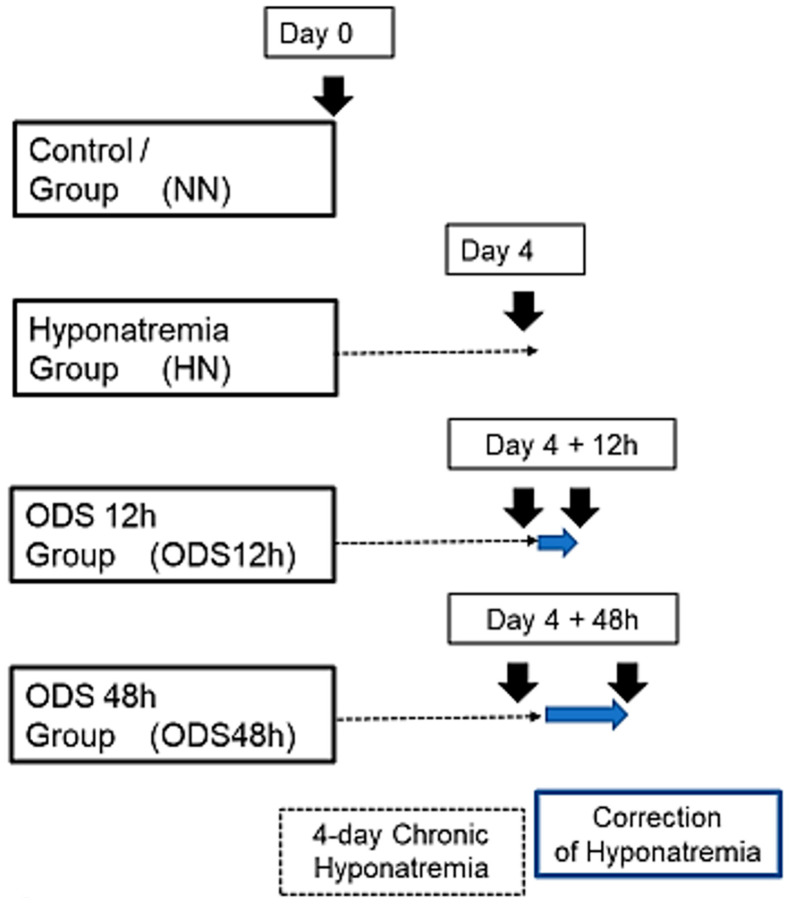
Experimental groups of mice undergoing chronic hyponatremia and correction of hyponatremia.

**Figure 2 ijms-24-16448-f002:**
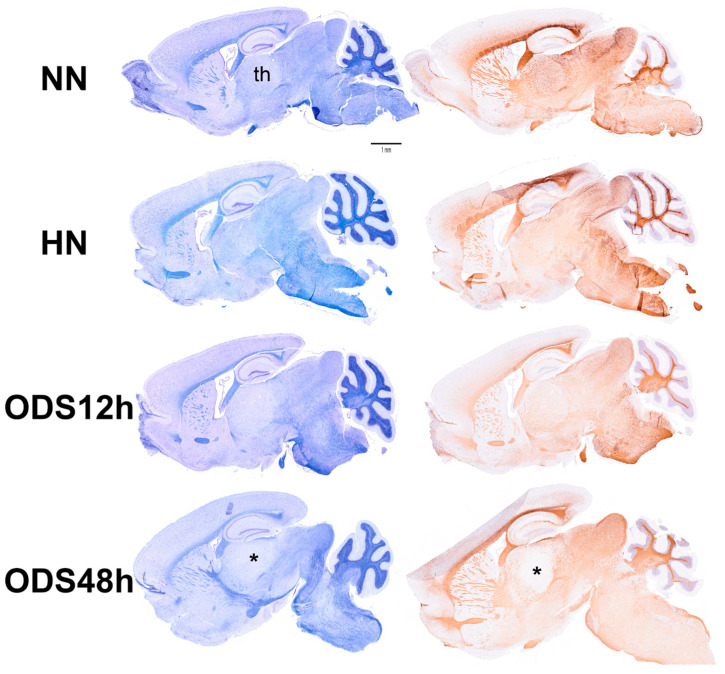
Murine brain parasagittal paraffin-sections stained with Eriochrome C (**left**) and immunostained against MBP protein (**right**). The section of the ODS48h brain where the thalamus (th) clearly reveals a poor contrast caused by the extrapontine regional demyelination. Scale bar represents 1 mm.

**Figure 3 ijms-24-16448-f003:**
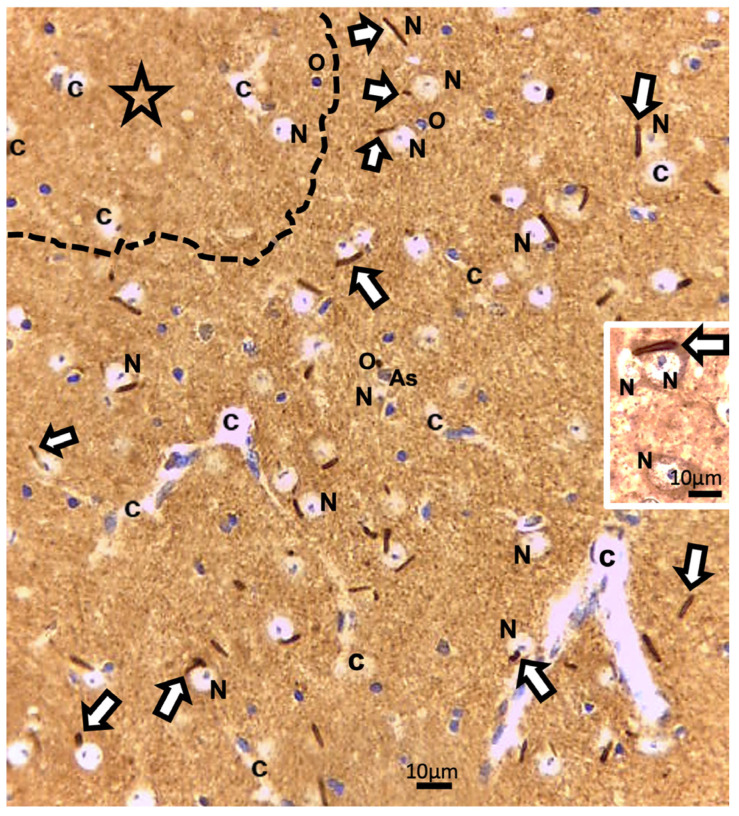
LM paraffin section of ODS12h ARL13b immunolabeled thalamus aspect. The upper left broken line delimitates a main part of the ODS damaged zone (star). Adjacent to and more distally, the thalamus field of view reveals many examples of some shafts or rod-shaped labeled axon initial segments (AIS), marked by white arrows among other cells and structures; As: Astrocyte; c: capillary; N: neuron; O: oligodendrocyte. Insert: an AIS can be recognized associating with joined thalamic neurons. Scales = 10 µm.

**Figure 4 ijms-24-16448-f004:**
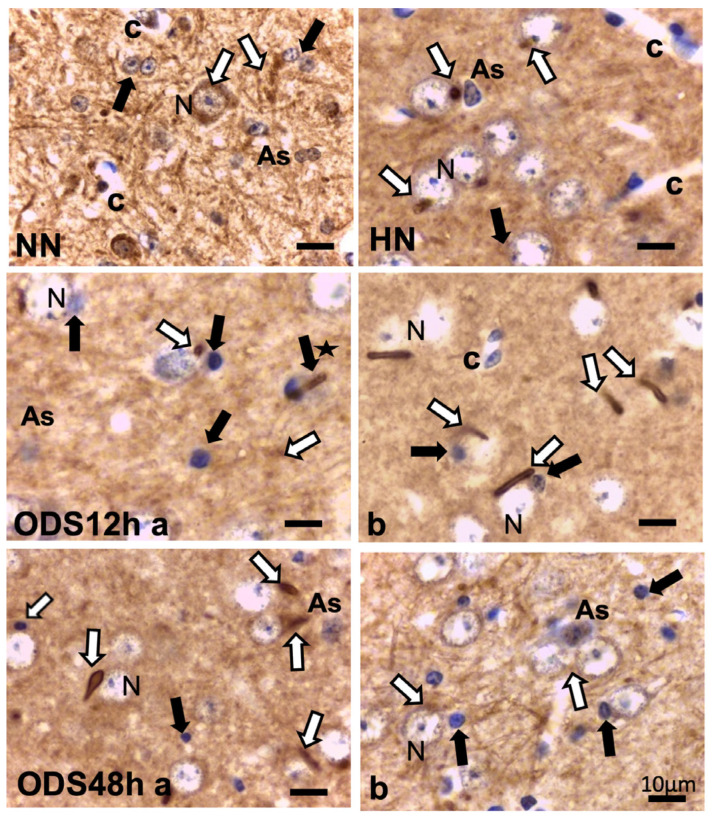
ARL13B immunolabeling of untreated (NN), chronic hyponatremic (HN,) and ODS murine thalamus VPL 12 h (ODS12h a and b) and 48 h (ODS48h a and b) post-treatment with hemalum counterstain. White arrows mark some examples of AIS, shaped from specks to rod-like straight or curved appendices in each frame of the pane that is associated with HN, ODS12h, and ODS48h thalamic nerve cell bodies (N) As: astrocyte; black arrows indicate some of the oligodendrocytes; black star indicates rod-like appendice emerging from oligodendrocyte; c: capillary. All the scales = 10 µm.

**Figure 5 ijms-24-16448-f005:**
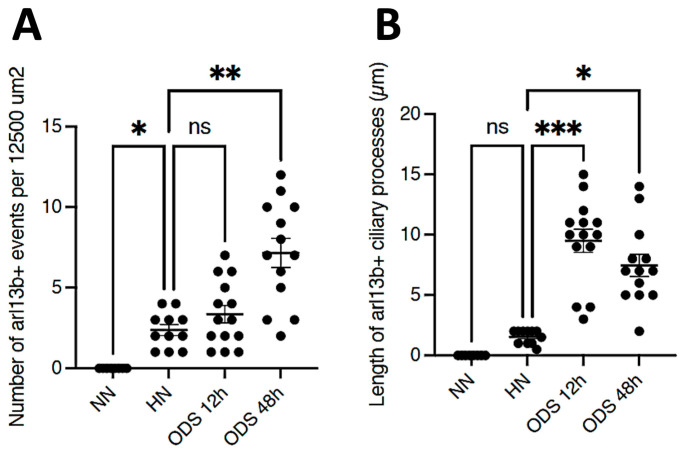
(**A**,**B**): Samples of counts and measurements of ARL13B AIS labeled from 12,500 µm^2^ LM fields of view compared between NN, HN, ODS12h, and ODS48h treatments. * *p* < 0.05, ** *p* < 0.01, *** *p* < 0.001, ns = not significant.

**Figure 6 ijms-24-16448-f006:**
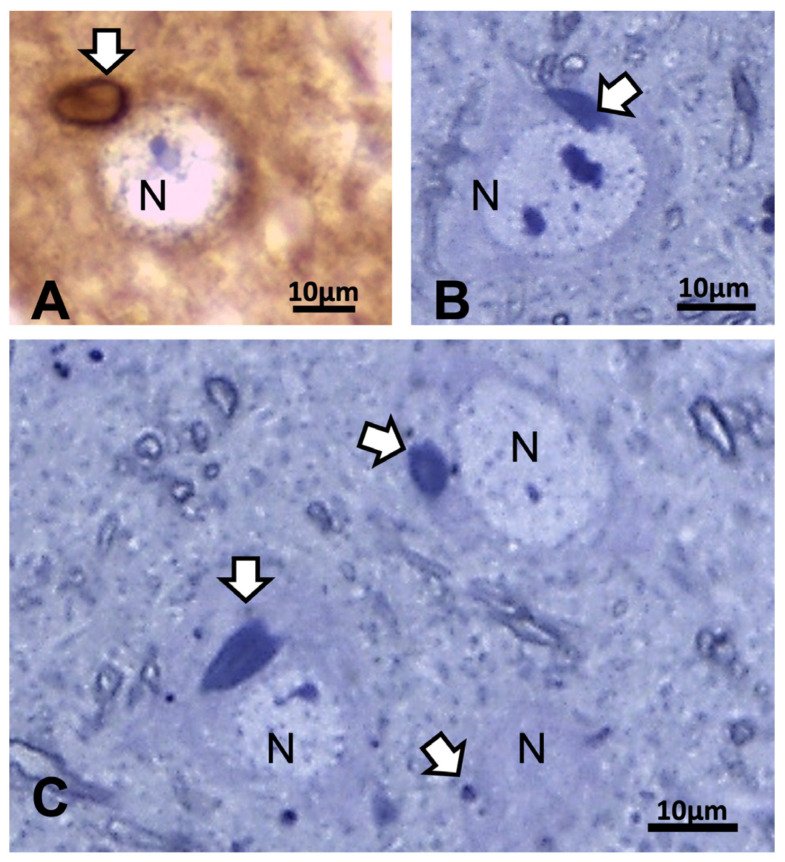
(**A**–**C**): LM aspects of 12 h ODS murine thalamus within 100 µm distance from the ODS damaged rim. (**A**): ARL13b immunolabeled depicting a small rod-shaped appendix (white arrow) to a nerve cell body (N) where hemalum has enhanced RNA in pale blue more than the DNA content of nucleolus. (**B**,**C**): One-µm thick epoxy section views of nerve cell bodies (N) where similar heavily basophilic shapes appeared issued from the narrow perikaryal zone; m: myelinated nerve bundles with diverse orientations. All the scales = 10 µm.

**Figure 7 ijms-24-16448-f007:**
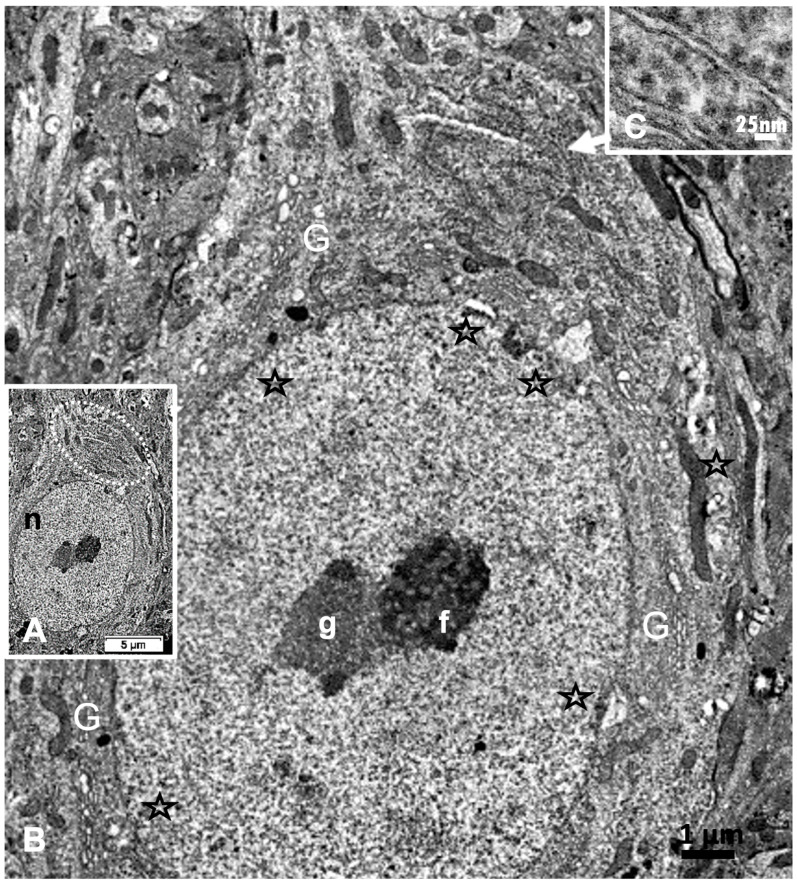
(**A**–**C**): **A**: TEM aspect of 12hODS murine thalamus within 100 µm distance from the ODS’ damaged edge revealing a neuron with a fine structure like those of Figure 6A–C with perikaryal content of an encircling Golgi apparatus (G) and especially containing a zone enriched with granulated content associated with membranes as RER (as revealed by (**B**)). The apex view region indicated by a white ellipsoid in A and marked by the white arrow had a perikaryon contrasted with granulations. As shown in an enlarged view of **C**, a neuroplasm congested by free and polyribonucleoproteins attached to endoplasmic reticulum; scale equals 25 nm. Note the stars adjacent to nuclear envelope pores marked adjacent neuroplasm damages. The nucleolus with numerous nucleolar organizer centers (NORs) made of dense and fine fibrils (f) and the large mass of RNA storage associated as granular center (g).

**Figure 8 ijms-24-16448-f008:**
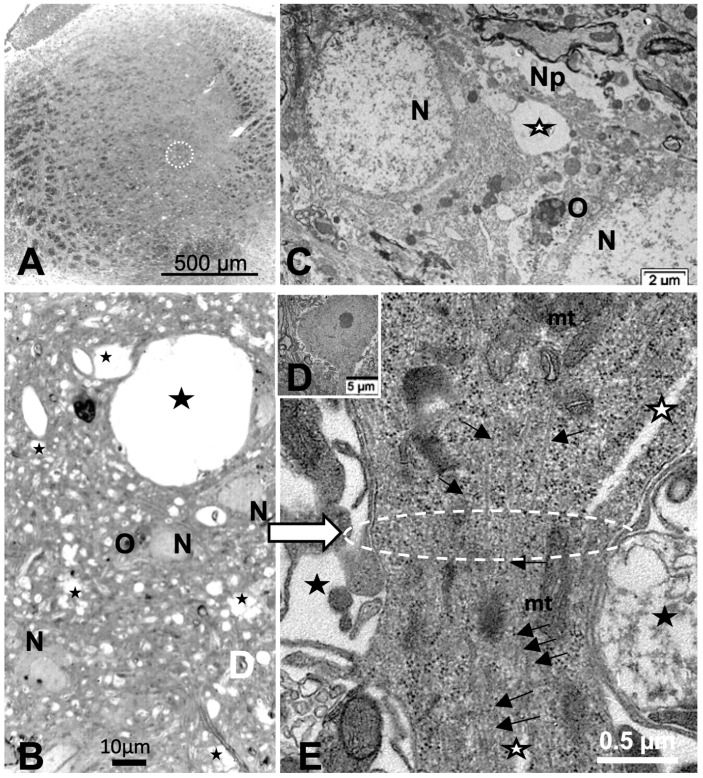
(**A**–**E**): (**A**): Parasagittal LM paraffin section part of ODS48h thalamic VPL region as viewed following blood–brain barrier immunoglobulin G permeability assessement [67], with the epicenter of the demyelinated region outlined; scale is 500 µm. (**B**): One-µm thick epoxy section, toluidine blue-stained of an area of the ODS demyelinated zone, shown filled with diverse sized vacuoles resulting from tissue degradations. M: microglial cell, N: neuron, O: oligodendrocyte; scale = 20 µm. (**C**–**E**): TEM aspects of similar spoiled neuropil (Np) ODS zone as in (**B**) where (**C**) showed damaged neurons (N) with diluted nucleoplasm and vacuoles (star) and lysosomes in perikaryal zones that included a necrotic oligodendrocyte satellite (O). (**D**): a nerve cell body with axon hillock, scale = 5 µm. (**E**): TEM enlarged aspect of (**D**) where the axon hillock (Ah) transition to the initial segment (AIS) is marked by a white ellipse, revealing microtubules among mitochondria (mt), and other associated organelles that became aligned in the latter, also underlined by neuroplasm gaps among the cytoskeleton (white stars). White arrow points a part of the AIS and black arrows indicate the local enrichment in neurotubules.

**Figure 9 ijms-24-16448-f009:**
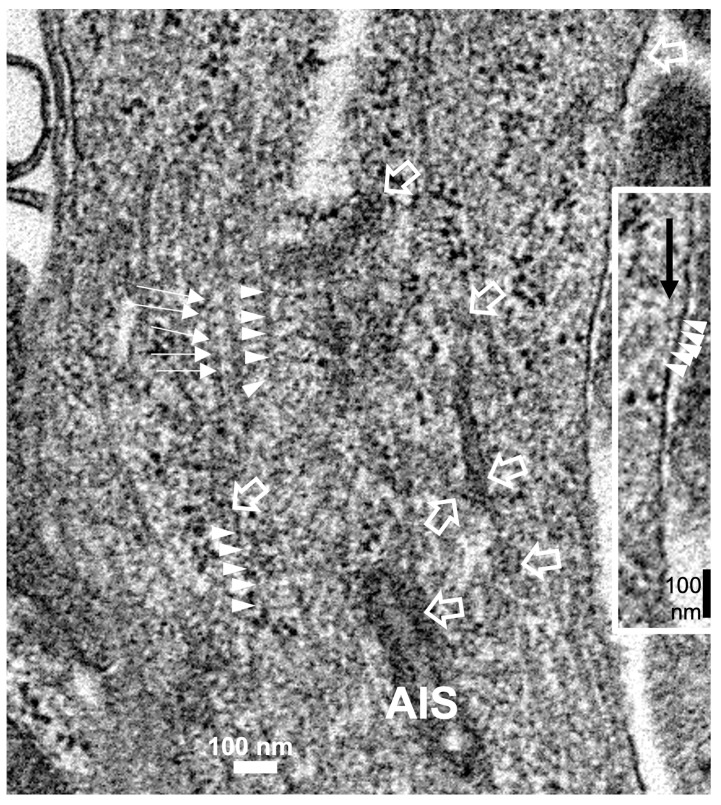
The enlarged view of the AIS segment, part in fine structure, shown in Figure 8E that illustrates the rich proteinaceous content of the thalamic cell body extension where the apparent blur-like view showed a rich microtubule core positioned into paraxial and parallel alignment and a tangent section profile of mitochondrion (mt) was shown and some cytoskeletal cross-linkages divulged their periodical architecture (small white arrows and arrowhead sets); in the profile, the sub neurolemma was underlined by a near distant delicate, neuroplasm protein arrangement (insert and white arrowheads); white open arrows indicate the swirled or twisted endoplasmic reticulum parts within the AIS crowded cytoskeleton. High contrast granules, 24–27 nm diameter, should be ribonucleoprotein strings.

**Figure 10 ijms-24-16448-f010:**
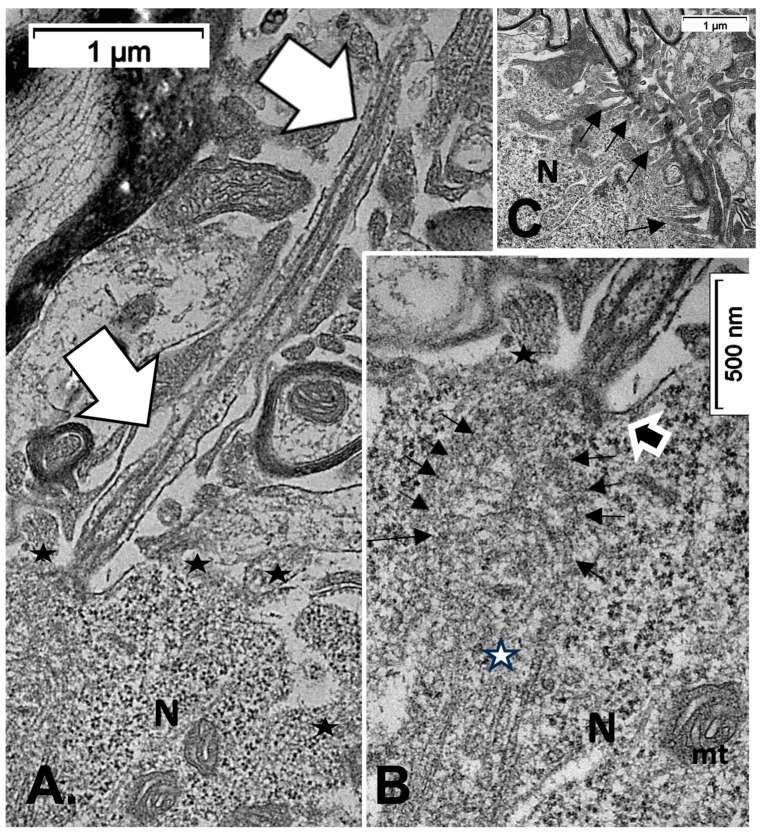
(**A**–**C**): TEM of an example of ODS48h resilient thalamic neuron (N), located in the ODS damaged zone, demonstrating a primary cilium (white arrows) among the degraded neuropil (**A**), and in its enlarged aspect (**B**) showing the narrow neck pocket (black arrow) whose basal aspect contains arrays of fibrillar structures that reach a hub-like area where a fascicle of microtubules also known as neurotubules issue from the perikaryon attained; these and other organelles (mt: mitochondrion) were shrouded by numerous ribonucleoproteins. Additionally, in both (**A**,**B**) illustrations, black stars mark diverse delicate lamellipodia, adjacent to the primary cilium. (**C**): filopodia (arrows) noticed adjacent to the primary cilium.

**Figure 11 ijms-24-16448-f011:**
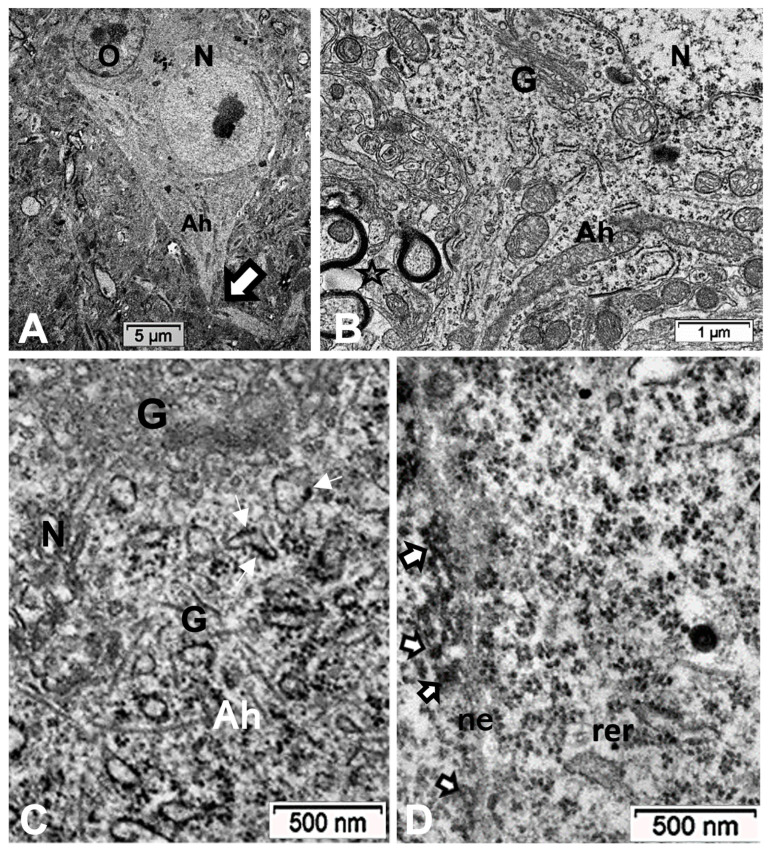
(**A**–**D**): TEM aspects of ODS48h thalamic nerve cell bodies located outside the outskirts of demyelination. A: Neuron (N) with its axon hillock (Ah) and twisted AIS (white arrow) among a neuropil where scattered, small intercellular spaces remain from ODS damage (stars) and a satellite oligodendrocyte (O). (**B**–**D**): Axon hillock regions reveal Golgi apparatus (G), mitochondria with winding endoplasm among free and attached polyribosomes. In (**B**), the AIS part with synaptic contact (white arrowhead) and adjacent astrocyte foot (As) are shown.

**Figure 12 ijms-24-16448-f012:**
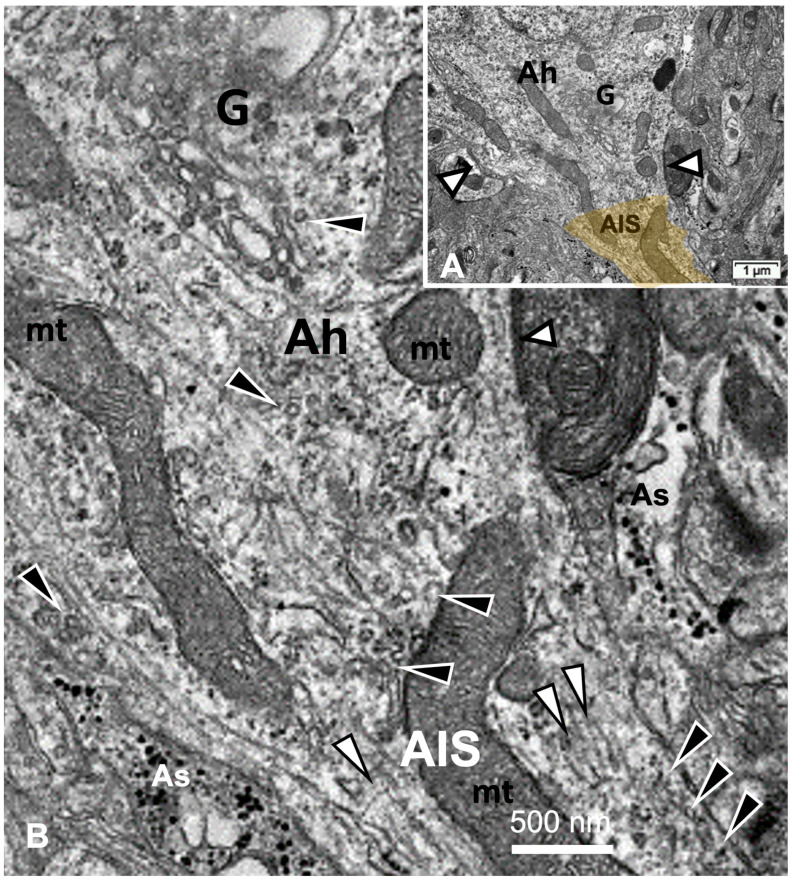
(**A**,**B**): TEM aspects of ODS48h thalamic neurons. (**A**): AIS early segment issued from axon hillock (Ah) marked by white arrows indicating synaptic connections. (**B**): Enlarged part of transition of axon hillock to AIS segment recognized by Golgi saccules (G) erupting with numerous vesicles and adjacent neurotubules that resolve from erratic orientation to a more paraxial to parallel funnel shape fascicle (white arrowheads), where vesicles (black arrowheads) located adjacent to neurolemma as cisternal organelles and along mitochondria (mt), typically without myelin, are surrounded by a few end-feet of astrocytes (As), recognized by glycogen particle contents.

**Figure 13 ijms-24-16448-f013:**
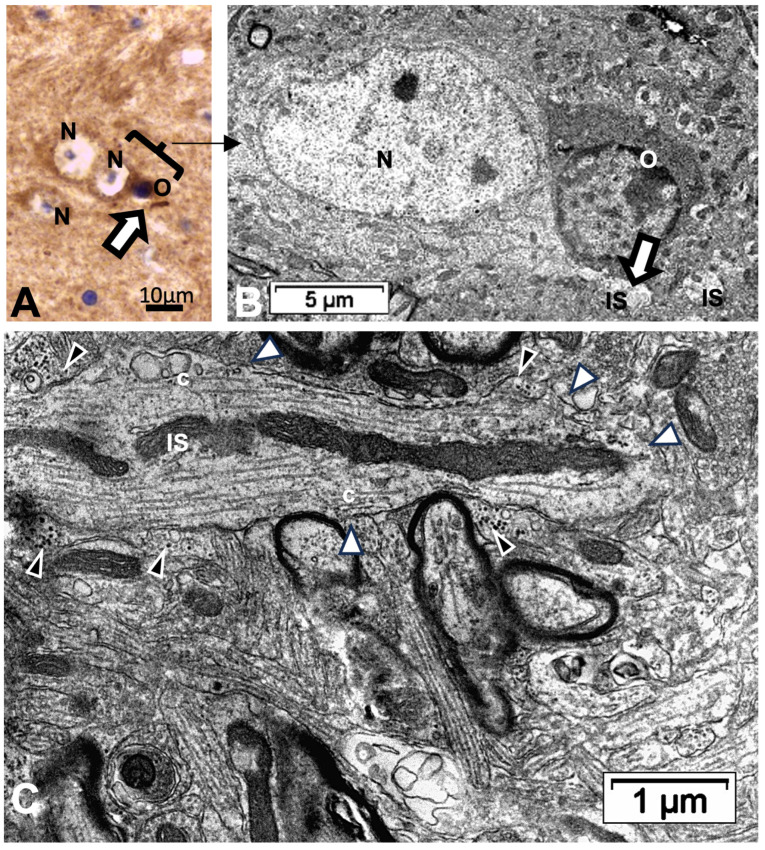
(**A**–**C**): Thalamic ODS48h nerve cell bodies and axon initial segment (IS). (**A**): LM view where two joined neurons have one on the right that displays an ARL13B labeled rod-like AIS structure (white arrow). N: neuron; O: oligodendrocyte. A bracket encompasses the TEM corresponding aspect in (**B**) with further sectioning level and enlargement in (**C**) if the axon IS among the neuropils, still depicting ODS damage remnants (stars); black arrowheads indicate astrocyte parts and white arrowheads mark synaptic contacts.

**Figure 14 ijms-24-16448-f014:**
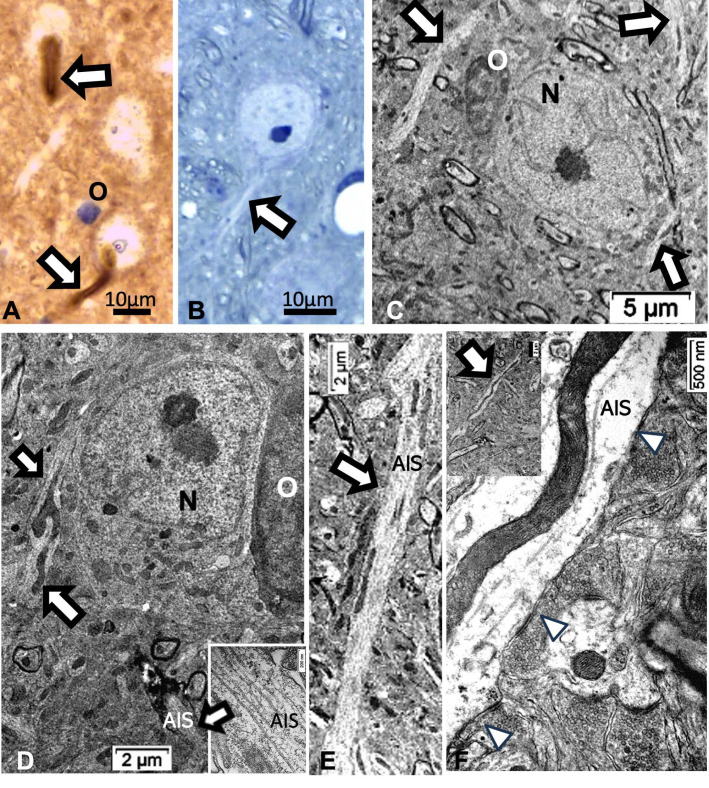
(**A**–**F**): Thalamic ODS48h nerve cell bodies and elongated axon initial segment (white arrows) aspects viewed with LM (**A**,**B**) and TEM (**C**–**F**). (**A**): ARL13B labeled AIS shown of two neurons as shaft and curved rod and in (**B**), from 1-µm thick epoxy section, toluidine blue-stained. (**C**,**D**): AIS from resilient neurons (N), including insert in (**D**), to show neurotubules’ bundle; O: oligodendrocyte. (**E**,**F**): Elongated AIS among the neuropils, containing huge mitochondria profiles and some synaptic contacts can be seen (white arrowheads).

**Figure 15 ijms-24-16448-f015:**
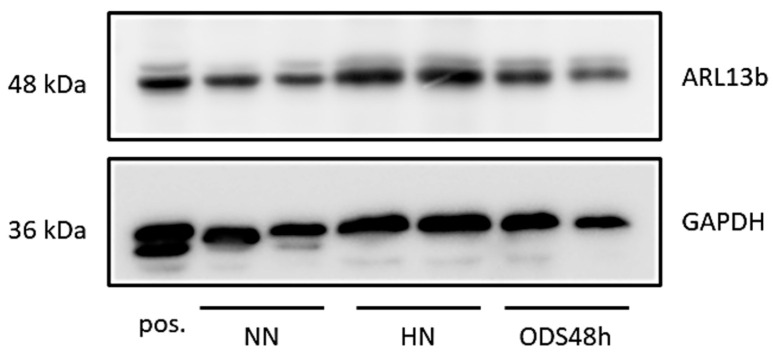
Western blots combining samples of NN, HN, and ODS48h thalamus where ARL13B protein is detected across all samples without significant variation; GAPDH is included as a loading control. A protein sample from mouse testis was used as positive (pos.) control.

**Table 1 ijms-24-16448-t001:** Evolution of natremia expressed in mEq/L along ODS protocol (mean ± SEM). One-way ANOVA followed by Dunn’s multiple comparisons test. *** *p* < 0.001 and **** *p* < 0.0001 compared to HN.

NN	HN	ODS24h	ODS48h
146.6 ± 2.1	118.2 ± 1.6	142.0 ± 5.9 ***	146.0 ± 5.1 ****

## Data Availability

Supporting data, as the original blots, are available as Appendix A.

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
