# Peer review of "Thalamic Neuron Resilience during Osmotic Demyelination Syndrome (ODS) Is Revealed by Primary Cilium Outgrowth and ADP-ribosylation factor-like protein 13B Labeling in Axon Initial Segment"

_ijms, 2023, doi:10.3390/ijms242216448_

Round 1
Reviewer 1 Report
Comments and Suggestions for Authors
In this study, the authors present the resilience of the thalamic neurons in osmotic demyelination syndrome through demonstration of primary cilium outgrowth and ARL13B labeling in the axon initial segment. The manuscript is very well written and can be accepted for publication in the IJMS journal. However, I have a few suggestions:
1- Please change the position of Material and Methods and put it after the Introduction section.
2- About the resilience of neurons, how about the changes that occur in the pontine neurons, Do these changes occur in the pontine neurons in addition to the thalamic neurons? please discuss this in the Discussion section.
Author Response
Reviewer #1:
In this study, the authors present the resilience of the thalamic neurons in osmotic demyelination syndrome through demonstration of primary cilium outgrowth and ARL13B labeling in the axon initial segment. The manuscript is very well written and can be accepted for publication in the IJMS journal. However, I have a few suggestions:
1- Please change the position of Material and Methods and put it after the Introduction section.
R/ As confirmed by journal’s associate editor, the correct layout format for IJMS is as it is originally prepared, which includes the following order: 1.Introduction, 2.Results, 3.Discussion, and 4.Materials and Methods. Even though it sounds more convenient as suggested by the reviewer, there is no need to change the position of the M&M section.
2- About the resilience of neurons, how about the changes that occur in the pontine neurons, Do these changes occur in the pontine neurons in addition to the thalamic neurons? please discuss this in the Discussion section.
R/ We thank the reviewer for the relevant remark. As described in Bouchat et al., Glia, 2018 (DOI: 10.1002/glia.23268), the pontine lesions were observed inconsistently across the ODS mice [in about 8 to 11 % of mice], while the thalamic nuclei were systematically affected. We therefore made the choice to analyze only the thalamic region. We added this limitation in the discussion (page 18, line 496).

Reviewer 2 Report
Comments and Suggestions for Authors
The manuscript presented describes a study exploring the impacts of chronic hyponatremia and its correction on murine models to understand the pathology and potential recovery mechanisms surrounding Osmotic Demyelinating Syndrome (ODS). Through a detailed examination using light microscopy, immunolabeling, and transmission electron microscopy, the authors highlight the changes in thalamic regions, particularly focusing on the resilience and potential repair mechanisms in neuronal structures following the onset of ODS. I believe the paper would be of great interest to the audience of IJMS.
Below are some specific suggestions to further improve the manuscript:
- The introduction is rich in background information. However, a more concise review of the current understanding of ODS pathophysiology and the gaps that this study aims to fill could make the objectives of this study clearer. It would be useful to include a clear statement of the hypothesis driving this study.
- The changes in serum [Na+] and their impact on the demyelination extent need to be statistically analyzed to establish significant trends and correlations.
- The manuscript dives deeply into microscopic observations but lacks a broader interpretation of what these findings mean in the context of ODS. It's important to provide a thorough analysis discussing the implications of the findings and how they advance the understanding of ODS.
- The role and significance of ARL13B protein in the recovery or resilience post-ODS could be elaborated on further. How does ARL13B labeling correlate with the structural and functional recovery in the thalamic region?
- The temporal analysis at 12h and 48h post-correction is crucial. Discussing how the observed changes evolve over time and their implications on the disease progression or recovery could add depth to the analysis.
Author Response
Reviewer #2:
The manuscript presented describes a study exploring the impacts of chronic hyponatremia and its correction on murine models to understand the pathology and potential recovery mechanisms surrounding Osmotic Demyelinating Syndrome (ODS). Through a detailed examination using light microscopy, immunolabeling, and transmission electron microscopy, the authors highlight the changes in thalamic regions, particularly focusing on the resilience and potential repair mechanisms in neuronal structures following the onset of ODS. I believe the paper would be of great interest to the audience of IJMS.
Below are some specific suggestions to further improve the manuscript:
- The introduction is rich in background information. However, a more concise review of the current understanding of ODS pathophysiology and the gaps that this study aims to fill could make the objectives of this study clearer. It would be useful to include a clear statement of the hypothesis driving this study.
R/ We thank the review for the constructive comment. We tried to clarify the rationale of the study in the last paragraph of the introduction (page 2, line 90).
- The changes in serum [Na+] and their impact on the demyelination extent need to be statistically analyzed to establish significant trends and correlations.
R/ We statistically analyzed the variation of sodium levels along the ODS protocol. The values are now provided as Table 1 (page 3). We did not intend to correlate the extent of demyelination with the magnitude of sodium correction.
- The manuscript dives deeply into microscopic observations but lacks a broader interpretation of what these findings mean in the context of ODS. It's important to provide a thorough analysis discussing the implications of the findings and how they advance the understanding of ODS.
R/ We partly agree with the reviewer. The paragraphs 3.2. and 3.3.3. already include some insights about the putative role of this primary cilium emergence.
- The role and significance of ARL13B protein in the recovery or resilience post-ODS could be elaborated on further. How does ARL13B labeling correlate with the structural and functional recovery in the thalamic region?
R/ This is a great suggestion from the reviewer. Behavior-based functional tests involving sensorimotor circuits and/or tracing of thalamocortical fibers could be investigated in the future.
- The temporal analysis at 12h and 48h post-correction is crucial. Discussing how the observed changes evolve over time and their implications on the disease progression or recovery could add depth to the analysis.
R/ We thank the review for the constructive comment. Longer time lapses post-ODS need indeed to be considered to evaluate the repair pattern of the neurons along with those of the neuropil, those of the astrocytes, oligodendrocytes and microglial cells, already studied in previous published data referred in the text that included protein and gene expression investigations as well as animal behavioral analysis. We added a sentence about the limitation(s) of the study in the discussion (page 22, line 690).

Reviewer 3 Report
Comments and Suggestions for Authors
This article investigated thalamic neuron resilience during Osmotic Demyelination Syndrome (ODS) in mice. ODS was induced by manipulating sodium levels to cause chronic hyponatremia followed by rapid correction. The most demyelinated regions were the thalamic ventral posterolateral and ventral posteromedial relay nuclei. In these areas, some resilient neuronal cell bodies extended axon hillocks and initial segments labeled with ARL13B, indicating transcription and translation activity. Neurons closer to the damaged ODS core had shorter axon segments, while those further away had longer segments. After 48 hours, some neurons had formed primary cilium, suggesting regeneration. Electron microscopy confirmed abundant ribonucleoproteins and organized cytoskeletal components in axon initial segments of resilient neurons. This shows thalamic neurons have some resilience and regenerative capacity after ODS damages.
This mouse model of ODS caused thalamic demyelination but some neurons showed resilience. Resilient neurons extended axon hillocks and initial segments marked by ARL13B protein. With time post-ODS, these segments grew longer and some neurons formed primary cilium, indicating regenerative potential. Ultrastructure analysis showed active transcription and translation with organized cytoskeletons in extending axons.
Comments
- Short time course - Only 12 and 48 hours post-ODS were evaluated. Longer time points could show more regeneration.
- One mouse strain - C57BL/6J mice were used. Testing other strains could improve generalizability.
- No behavioral tests - Functional improvements were not directly assessed. Adding tests like gait analysis or sensory thresholds could help.
- No blinded analysis - Researchers were not blinded to experimental conditions. This can introduce bias. Blinding should be implemented.
- No negative control - An additional control without ARL13B labeling could have confirmed staining specificity.
- Limited endpoints - More markers of degeneration and regeneration could have been measured, like NF, GAP43, etc.
- Mechanism not studied - The mechanism of regeneration was not elucidated. Signaling pathways or epigenetic changes should be explored.
- No treatment tested - Compounds or rehab methods that could enhance regeneration were not tried. Testing interventions could have clinical implications.
Author Response
Reviewer #3:
This article investigated thalamic neuron resilience during Osmotic Demyelination Syndrome (ODS) in mice. ODS was induced by manipulating sodium levels to cause chronic hyponatremia followed by rapid correction. The most demyelinated regions were the thalamic ventral posterolateral and ventral posteromedial relay nuclei. In these areas, some resilient neuronal cell bodies extended axon hillocks and initial segments labeled with ARL13B, indicating transcription and translation activity. Neurons closer to the damaged ODS core had shorter axon segments, while those further away had longer segments. After 48 hours, some neurons had formed primary cilium, suggesting regeneration. Electron microscopy confirmed abundant ribonucleoproteins and organized cytoskeletal components in axon initial segments of resilient neurons. This shows thalamic neurons have some resilience and regenerative capacity after ODS damages.
This mouse model of ODS caused thalamic demyelination but some neurons showed resilience. Resilient neurons extended axon hillocks and initial segments marked by ARL13B protein. With time post-ODS, these segments grew longer and some neurons formed primary cilium, indicating regenerative potential. Ultrastructure analysis showed active transcription and translation with organized cytoskeletons in extending axons.
Comments
- Short time course - Only 12 and 48 hours post-ODS were evaluated. Longer time points could show more regeneration.
R/ We thank the review for the constructive comment. Longer time lapses post-ODS need indeed to be considered to evaluate the repair pattern of the neurons along with those of the neuropil, those of the astrocytes, oligodendrocytes and microglial cells, already studied in previous published data referred in the text that included protein and gene expression investigations as well as animal behavioral analysis. We added a sentence about the limitation(s) of the study in the discussion (page 22, line 690).
- One mouse strain - C57BL/6J mice were used. Testing other strains could improve generalizability.
R/ Since we have been working on the experimental models of ODS, this is an investigation that we have always thought about, but we were lacking time.
- No behavioral tests - Functional improvements were not directly assessed. Adding tests like gait analysis or sensory thresholds could help.
R/ This is a great suggestion from the reviewer. Behavioral tests involving sensorimotor circuits will be implemented in the future.
- No blinded analysis - Researchers were not blinded to experimental conditions. This can introduce bias. Blinding should be implemented.
R/ We agree with the reviewers that the study could have been better designed with full blinding of the investigator. We specified this limitation in the M&M section (page 23, line 764).
- No negative control - An additional control without ARL13B labeling could have confirmed staining specificity.
R/ ARL13B technical and biological controls for immunolabeling specificity were verified on non-nervous tissues, i.e., the seminiferous tubules and the epididymis epithelia where flagella and stereocilia appendages dynamic turnover necessitated constant transport of subcellular components to maintain their functions. These new data were added as supplementary material Figure S3.
- Limited endpoints - More markers of degeneration and regeneration could have been measured, like NF, GAP43, etc.
R/ This is a great suggestion and will be investigated in the future, in particular the full characterization of the primary cilium components with additional markers and its fate on longer periods post-ODS.
- Mechanism not studied - The mechanism of regeneration was not elucidated. Signaling pathways or epigenetic changes should be explored.
R/ We agree with the reviewers that it could have been tested but this was not the objective of the current study.
- No treatment tested - Compounds or rehab methods that could enhance regeneration were not tried. Testing interventions could have clinical implications.
R/ We agree with the reviewers that it could have been tested but this was not the objective of the current study.
